# Investigation of the Prognostic Role of Carbonic Anhydrase 9 (CAIX) of the Cellular mRNA/Protein Level or Soluble CAIX Protein in Patients with Oral Squamous Cell Carcinoma

**DOI:** 10.3390/ijms20020375

**Published:** 2019-01-16

**Authors:** Alexander W. Eckert, Susanne Horter, Daniel Bethmann, Johanna Kotrba, Tom Kaune, Swetlana Rot, Matthias Bache, Udo Bilkenroth, Waldemar Reich, Thomas Greither, Claudia Wickenhauser, Dirk Vordermark, Helge Taubert, Matthias Kappler

**Affiliations:** 1Department of Oral and Maxillofacial Plastic Surgery, Martin Luther University Halle-Wittenberg, 06120 Halle (Saale), Germany; susanne.horter@uk-halle.de (S.H.); johanna.kotrba@med.ovgu.de (J.K.); tom.kaune@uk-halle.de (T.K.); swetlana.rot@medizin.uni-halle.de (S.R.); waldemar.reich@uk-halle.de (W.R.); matthias.kappler@uk-halle.de (M.K.); 2Department of Radiotherapy, Martin Luther University Halle-Wittenberg, 06120 Halle (Saale); Germany; matthias.bache@uk-halle.de (M.B.); dirk.vordermark@uk-halle.de (D.V.); 3Institute of Pathology, Martin Luther University Halle-Wittenberg, 06120 Halle (Saale), Germany; daniel.bethmann@uk-halle.de (D.B.); claudia.wickenhauser@uk-halle.de (C.W.); 4Institute of Pathology, 06295 Eisleben, Germany; udobilkenroth@gmx.de; 5Center for Reproductive Medicine and Andrology, Martin Luther University Halle-Wittenberg, 06120 Halle (Saale), Germany; thomas.greither@medizin.uni-halle.de; 6Clinic of Urology and Pediatric Urology, FA University Hospital Erlangen-Nürnberg, 91054 Erlangen, Germany; Helge.Taubert@uk-erlangen.de

**Keywords:** CA9, carbonic anhydrase IX, survival, hypoxia, OSCC, CAIX

## Abstract

Carbonic anhydrase 9 (CAIX) is an important protein that stabilizes the extracellular pH value and is transcriptionally regulated by hypoxia-inducible factor 1 (HIF1), but more stable than HIF1α. Here we show a comparative study that examines the prognostic value of CA9 mRNA, CAIX protein of tumor cells and secreted CAIX protein for oral squamous cell carcinoma (OSCC) patients. Tumor samples from 72 OSCC patients and 24 samples of normal tissue were analyzed for CA9 mRNA levels. A total of 158 OSCC samples were stained for CAIX by immunohistochemistry and 89 blood serum samples were analyzed by ELISA for soluble CAIX protein content. Survival analyses were performed by Kaplan–Meier and Cox’s regression analysis to estimate the prognostic effect of CA9/CAIX in OSCC patients. The CA9 mRNA and CAIX protein levels of tumor cells correlated with each other, but not with those of the secreted CAIX protein level of the blood of patients. ROC curves showed a significant (*p* < 0.001) higher mRNA-level of CA9 in OSCC samples than in adjacent normal tissue. Cox’s regression analysis revealed an increased risk (i) of death for patients with a high CA9 mRNA level (RR = 2.2; *p* = 0.02), (ii) of locoregional recurrence (RR = 3.2; *p* = 0.036) at higher CA9 mRNA levels and (iii) of death at high CAIX protein level in their tumors (RR = 1.7; *p* = 0.066) and especially for patients with advanced T4-tumors (RR = 2.0; *p* = 0.04). However, the secreted CAIX protein level was only as a trend associated with prognosis in OSCC (RR = 2.2; *p* = 0.066). CA9/CAIX is an independent prognostic factor for OSCC patients and therefore a potential therapeutic target.

## 1. Introduction

Oral cancer is the 15th most common cancer worldwide [1]. Oral squamous cell carcinoma (OSCC) is the major malignancy of all oral cancers [2,3]. However, the prognosis of OSCC has stagnated over the last decades [3]. Therefore, new prognostic molecular parameters are needed to characterize the biology of OSCC and to enable an individual therapy concept. The history of determining additive prognostic parameters in OSCC dates back to 2003. Schliephake had analyzed the literature from five years (1997–2002) and included 169 articles in his review [4]. The complex field of tumor markers in OSCC has been classified into four groups according to their function: (i) enhancement of tumor growth: cell cycle acceleration and proliferation and markers such as epidermal growth factor receptor (EGFR), Ki67 or heat shock protein 70 (ii) tumor suppression and anti-tumor defense: Immune response and apoptosis such as pRB, p16 or p53, (iii) angiogenesis such as vascular endothelial growth factor receptor (VEGFR) or NOS2, (iv) tumor invasion and metastatic potential: adhesion molecules and matrix degradation such as matrix-metallo-proteases, cathepsins, integrins or desmoplakin/plakoglobin [4]. He concluded that the finding on tumor invasion and metastatic potential markers appeared too early to predict their prognostic value [4]. Only a few years later, Lothaire postulated that some associations between molecular markers and invasiveness, aggressiveness, the degree of differentiation, and tumor stage were found, but only a few clinical studies have shown any impact on prognosis in OSCC [5]. The most recent reviews partly refer to new biomarkers in OSCC [6,7,8]. 

New interesting approaches to estimating the individual prognosis of OSCC patients are the analysis of the immune profile of OSCC [9,10,11], the analysis of long non-coding RNAs [12,13], and miRNAs, e.g., from the serum or saliva of patients as a liquid biopsy [14,15,16,17,18,19], metabolic products [20] or even cell-free DNA [21] also using new technologies such as the proteomic approach or microarray [22,23,24]. 

An important prognostic feature in OSCC is tumor hypoxia. Hypoxia can be demonstrated by the overexpression of the α-subunit of the transcription factor hypoxia-inducible factor 1 (HIF-1) [25]. With few exceptions [26,27], this factor HIF-1α is indeed essentially detectable and active under hypoxic conditions. However, the stability of HIF-1α is very limited and a more stable molecular parameter that could be used as a surrogate marker for hypoxia is necessary. Such a marker could be carbonic anhydrase 9 (CAIX), which is a specific transcriptional target gene of HIF-1, but more stable than HIF-1α [28,29]. Carbonic anhydrase 9 (CAIX), as a member of the family of metalloenzymes, plays an important role in stabilizing the extracellular pH value by the hydratization of carbon dioxide [30] which influences tumor progression of solid cancers [31]. 

Regarding the prognostic relevance of CAIX in OSCC, there are different results [32], however, an association of the expression level of CAIX in OSCC with metastatic processes has been described in several reports [33,34,35,36]. Yang et al. found CA9 mRNA levels as prognostically relevant using datasets from The Cancer Genome Atlas (TCGA) of patients with oral cancer [35]. Peterle et al. [37] and Brockton et al. [32] described the CAIX protein level to be prognostically relevant to OSCC. Lin et al. found a worse outcome of OSCC patients when lipocalin 2 (LCN2) was weak but CAIX was strongly expressed [33].

However, CAIX has also been described as a prognostically relevant biomarker in a wide range of different tumors e.g., breast cancer (Hazard ratio (HR = 2.2)) [38], head and neck cancer (HR = 1.8) [39], lung cancer (HR = 3.3) [40], soft tissue sarcoma (HR = 2.2) [41] and liver carcinoma (HR = 1.9) [42]. A recent overview of the meta-analysis of 147 independent studies evaluating the prognostic effects of CAIX on different tumors was published by van Kuijk et al. [31]. 

The potential effect of CAIX on processes of epithelial mesenchymal transition (EMT) and metastasis in OSCC could also provide a new therapeutic approach for OSCC at directly targeting CAIX [43].

The aim of this study was to analyze the expression of carbonic anhydrase 9 in tumor tissue on mRNA (CA9) or the protein (CAIX) level and the level of secreted CAIX in the blood serum of OSCC patients to determine its potential as a prognostic biomarker and for future therapeutic stratification.

The novelty of this analysis is the direct comparison of the prognostic value of, on the one hand, CA9 mRNA and CAIX protein of the tumor tissue and, on the other hand, the secreted CAIX protein from the blood serum of OSCC patients of the same cohort.

## 2. Results

### 2.1. Analysis of the mRNA Level of CA9 in Tumor Tissue Compared to Tumor-Associated Normal Tissue 

The CA9 mRNA levels of 72 OSCC samples as well as of 24 tumor adjacent tissue samples were measured and normalized to the RNA polymerase II (RPII) mRNA level. The median expression in the tumor tissue was 100 (mean 742.8), while the median expression in tumor-associated normal tissue samples was 4.4 (mean 176) (Figure 1). The CA9 mRNA level in tumor tissue samples was significantly higher compared to tumor-associated normal tissue samples in this study cohort. The diagnostic impact of the CA9 transcription level was analyzed by a ROC curve. In this way, the CA9 mRNA content can be used as a significant parameter (*p* < 0.001) for the differentiation of tumors from normal tissue with an AUC (area under the curve) of 0.82. The Youden point was determined as 15.9. Using the Mann–Whitney U-test this significant result of the ROC curve was confirmed (*p* < 0.001).

### 2.2. Association of the CA9 mRNA Level with the Survival of OSCC Patients 

The CA9 mRNA level of 72 OSCC samples was divided into two groups according to the CA9 mRNA concentration. As an optimal cut-off point, we defined 70% of the tumor samples (n = 51; CA9 mRNA level < 246, range: 0.63–246) with the lowest CA9 level as a low CA9 mRNA level, while the remaining 30% (n = 21) of the tumor samples were defined as overexpressing CA9 mRNA of ≥246.1 (range: 246.1–33779). 

Patients with high levels of CA9 mRNA in the tumor tissue had a median overall survival of 11 months post-diagnosis compared to patients with low CA9 mRNA level who had a median overall survival of 42 months (Kaplan–Meier–analysis (*p* = 0.038)). The multivariate Cox’s regression hazard analysis (adjusted to the patient’s tumor stage, lymph node status (N-stage) and tumor grade), showed an increased risk of death (RR = 2.2; *p* = 0.02) for patients with a high CA9 mRNA level in their tumors as compared to patients with a low CA9 mRNA level (Table 1, Figure 2).

In addition, multivariate Cox’s regression analysis (adjusted to the patient’s tumor stage, lymph node status (N-stage) and tumor grade) revealed that a high CA9 mRNA level in the tumor is associated with an increased risk of locoregional recurrence. Patients with high levels of CA9 mRNA had an increased relative risk for a relapse of RR = 3.2 (*p* = 0.036) compared to patients with a low CA9 mRNA level in their tumors (Figure 3). 

### 2.3. Association of the CAIX Protein Level with the Survival of OSCC Patients 

Tumor tissues from 158 OSCC patients were stained specifically for CAIX by IHC. The clinical and pathological data are summarized in Table 1. The analysis showed that 134 (85%) patients had a negative, weak or moderate CAIX protein content (immunoreactive score (IRS) 0–6) in the tumor and only 24 (15%) of the patients had a strong expression for CAIX (IRS 8–12) in the tumor tissue. The typical immunohistochemical staining for CAIX is shown in Figure 4.

Patients with high levels of CAIX protein in the tumor tissue had a median overall survival of 17 months post-diagnosis compared to patients with a low CAIX protein level who had a median overall survival of 46 months (Kaplan–Meier–analysis (*p* = 0.027)). In the multivariate Cox’s regression analysis (adjusted to patient’s tumor stage, lymph node status (N-stage) and tumor grade), a high CAIX protein level in the tumor showed a trend of significance for an increased risk of death (RR = 1.7; *p* = 0.066) compared to patients with a low CAIX protein level in their tumor (Table 2, Figure 5). 

Patients diagnosed with a stage T4 tumor with high levels of CAIX protein in the tumor tissue had a median overall survival of 6 months post-diagnosis compared to patients with a low CAIX protein level who had a median overall survival of 20 months (Kaplan–Meier–analysis (*p* = 0.15)). In addition, in the multivariate Cox’s regression hazard analysis for locally advanced tumors, this association was more pronounced. For patients diagnosed with a stage T4 tumor, the protein expression of CAIX showed a statistically significant 2.0- fold increased risk of death compared to patients with T4 tumors (*p* = 0.04, Figure 6). 

### 2.4. Association of the CAIX Protein Level in the Blood Serum of OSCC Patients with Survival 

CAIX can be detected not only as a localized protein on the cell surface, but also as a soluble protein in the blood serum. We examined the expression of CAIX in 89 blood serum samples from OSCC patients by ELISA. The mean expression level in the cohort of OSCC patients was 69 pg CAIX per ml blood serum (median 112 pg/mL; range 16.5–1117). The mean expression level in the cohort of 43 healthy blood donors was 87 pg CAIX per ml blood (median 70 pg/mL; range 15.8–281.1).

The OSCC cohort was divided into three groups according to their protein content in the blood serum of the patients (1) the group (n = 43; 48%) with a low CAIX level (16–64 pg/mL) (2) the group (n = 32; 36%) with a moderate-high CAIX level (≥ 65–139 pg/mL), and (3) the group (n = 14; 16%) with a very high CAIX level (≥140–1117 pg/mL). 

Patients with the highest levels of CAIX serum protein had a median overall survival of 19 months post-diagnosis compared to patients with low CAIX serum protein level who had a median overall survival of 44 months (Kaplan–Meier–analysis (*p* = 0.03)). The multivariate Cox’s regression analysis revealed that a very high CAIX level (≥ 140–1117 pg/mL) was associated with a tendency to a significantly increased risk of death (*p* = 0.066, RR = 2.2 95%CI: 0.95–5.05) compared to the reference (low CAIX level (16–64 pg/mL)) (Figure 7).

### 2.5. Correlation of the CA9 mRNA, CAIX Protein Expression and CAIX Protein Level in the Blood 

In a bivariate Spearman’s rank correlation, we found a significant correlation of CA9 mRNA with CAIX protein (IHC) levels (*p* < 0.001). However, the CA9 mRNA did not correlate with the CAIX protein level in the blood serum of the same patients (*p* = 0.49) (Table 3). However, the expression of CAIX protein in the tumor cells also did not correlate with the level of CAIX in the blood serum (*p* = 0.11).

### 2.6. Correlation of CA9 mRNA, CAIX Protein Expression and CAIX Protein Levels in the Blood Serum with Other Tumor-Related Markers

The CA9 mRNA level was also correlated with the mRNA expression of other HIF1 target genes in the investigated tumor samples such as glucose transporter 1 (Glut1≙ SLC2A1), glyceraldehyde-3-phosphate dehydrogenase (GAPDH), VEGFa and with miRNA-210 expression (Table 3). In addition, the CA9 mRNA level was correlated with the mRNA expression of survivin and keratin 13 (KRT13) (Table 3).

The CAIX protein (IHC) level of the tumor cells was correlated, as expected, with the HIF1α-protein (IHC) expression (*p* = 0.015) in the same tumor samples (Table 3). 

However, the soluble CAIX protein level measured in patients’ blood serum correlated with the level of miRNA-210 measured in the same blood serum but not with the miRNA-210 level in the same patients’ tumor. In addition, there appears to be a correlation of soluble CAIX, EGFR and HER2 levels (Table 3). 

## 3. Discussion

The identification of prognostic parameters for OSCC is a necessary approach to categorize the molecular and biological characteristics of individual tumors. A systematic categorization started with a review by Schliephake [4] and was continued by other authors [5] who gave a general overview of the potential markers in OSCC and outlined important cornerstones of future research questions.

One important feature of tumor biology is tumor hypoxia and, in particular, the activity of the transcription factor hypoxia-inducible factor (HIF). The prognostic relevance of the α-subunit of HIF1 in OSCC patients has already been demonstrated by our group [25,44,45] and others [46,47]. Zhou and co-workers showed an association of overexpression of HIF-1α with tumor size, tumor stage, lymph node metastasis, and overall survival. HIF-1α is suggested as an independent prognostic marker in patients with OSCC [47]. However, the main problem of HIF-1α is its very short half-life time of only a few minutes [48].

The complex HIF system regulates the expression of many genes that contribute to metabolic reprogramming, extracellular matrix remodeling, epithelial-to-mesenchymal transition, motility, invasion, metastasis, and many other adaptive strategies [49]. Among all HIF-1α mediated proteins, CAIX (a carbonic anhydrase), is one of the most promising tumor-associated proteins in the carcinogenesis process of OSCC [50]. However, it can also be regulated by other factors such as EGFR [30,51]. In addition, even an association of EGFR and HIF-1α was postulated [52,53]. This means that a HIF1 activity is still affected by an EGFR-mediated activation of CAIX. Most importantly, CAIX is a very stable protein with a half-life time of about 40 h [30]. This is particularly important for its use as a prognostic factor in order to carry out a stable and reproducible read out analysis.

In this study, we detected a higher CA9 mRNA level in the tumor tissue compared to normal tumor-adjacent tissue what confirms results from Yang et al. [35]. They investigated the CA9 mRNA level using datasets from The Cancer Genome Atlas (TCGA) of patients with oral cancer [35]. Furthermore, we found that those patients with a high tumor CA9 mRNA level (30% of all patients (n = 21)) had a significantly worse prognosis (Figure 2). Another study in OSCC described the effect of CA9 mRNA, but it was isolated and measured out of the peripheral blood of patients with oral leukoplakia [54]. These authors found a significantly higher CA9 mRNA level in the blood of two tumor patients [54]. In our analyses, we also found an association of the high CA9 mRNA level with an increased risk of an earlier locoregional recurrence (Figure 3) and identified CA9 mRNA as an independent prognostic factor for OSCC. 

The CAIX protein content has already been well studied, especially for OSCC patients [50] or for other tumors [31]. Peterle et al. found in a multivariate analysis of 26 OSCC tissue samples that the CAIX protein content was significantly associated with the risk of disease-related death (RR = 2.8; *p* = 0.045) [37]. We could show in this work that high CAIX expression (IRS 8–12) was associated with a trend of an increased risk of early death (RR = 1.7; *p* = 0.066). This effect is significant in the T4 tumor group as compared to the T1 group (RR = 2.0; *p* = 0.04 (Figure 5 and Figure 6)). Brockton et al. found an association with the disease-specific survival (HR = 1.99) in a cohort of 168 OSCC patients for the patients with the 25% highest CAIX protein level [32]. However, the authors described an optimal cut-off point for defining a tumor as a highly expressing CAIX tissue to be as high as 20% (33/168 patients) of all investigated patients (HR, 2.45; 95% CIs, 1.40–4.31) [32]. We have defined an optimal cut-off point to be as 15% of our cohort as highly expressing CAIX tumors. Altogether, it appears that only a few tumor patients, those with the highest CAIX protein levels, have a significantly worse prognosis. 

The combination of two markers such as CAIX and its transcription factor HIF-1α could be one possibility to verify the prognostic influence of hypoxic markers. In two independent cohorts, we showed that the high expression of CAIX and HIF-1α is of prognostic significance [44] and the best prognosis for OSCC patients in an earlier study of our group was actually found for low HIF1a ⁄CAIX protein expressing tumor patients [55]. Roh and coworkers, like us, found a correlation of CAIX and HIF-1α protein content, but these authors failed to show any prognostic relevance of either CAIX or HIF-1α in the investigated cohort of 43 T2-staged oral tongue tumor patients [56]. 

In this context, it was often described that CAIX could have a direct impact on EMT and metastasis in OSCC [31,33,35,36,57,58]. The influence of CAIX on EMT is often explained by CAIX’s effect on the pH value and the acidification of the tumor microenvironments [30,59,60,61,62]. In our analysis, we do not find a correlation between the CA9/CAIX level and, e.g., positivity of the lymph node status (Table 1, Table 2). On the other hand, however, the expression of CAIX necessarily required the activity of the transcription factor HIF1. However, HIF1 has a tremendous number of target genes, including *CA9*, that are definitively associated with EMT and metastasis, such as, e.g., LOX and P4HA1 [63,64]. It appears that the dominating effect of HIF1 [65] on various aspects of tumor biology may sometimes overshadow the effect of CAIX. The specific effects of CAIX on the metastasis processes should, therefore, be investigated in a HIF1 knock out cell line system. 

CAIX and HIF1 are, of course, important for stabilizing the intracellular and extracellular pH of the tumor cell and are key proteins of carcinogenesis [26,66]. In particular, the role of CAIX could be beyond its hypoxia-related function. It neutralizes the tumor-specific metabolism and, especially, the waste product of glutaminolysis ammonia, which occurs in the extracellular matrix [26,27,30,60,67,68]. It might be that the metabolic imbalances in tumor cells regulated by HIF’s appear to be more important for tumor progression and that CAIX and CAXII are important markers to counteract a malfunctioning tumor balance. It has been described that hypoxic/acidic tumor cells have an extracellular pH value in the range of 6.5–7.1, which is realized and controlled by CAIX [30,62]. This acidic tumor-specific environment, realized by CAIX, is, in our estimation, necessary for detoxification of the toxic waste product ammonia which is produced in large quantities by tumor-specific catabolism of the amino acid and nitrogen source glutamine. Then the toxic gas ammonia would be almost completely neutralized to ammonium at a pH value below 7.0 [26,27,69]. This could be realized by a HIF1-induced CAIX activity which changes the microenvironments to allow for the better survival of the tumor cell, but to the detriment of the patient. In this regard, the therapeutic approach against the tumor marker HIF1 and CAIX [43,70] is promising and could help to disrupt sensitively the tumor microenvironment, which could be beneficial for the patients.

Another way to analyze elegantly the biomarker CAIX is to determine its level directly in patient’s blood as a liquid biopsy. The ectodomain of the CAIX protein can be released into the extracellular space and into the blood by metalloprotease mediated processes [71]. We studied 89 blood samples of OSCC and identified 14 patients (16%) with very high blood levels of CAIX protein (140–1117 pg/mL). However, the solubilized CAIX level (*p* = 0.066, RR = 2.2 95%CI: 0.95–5.05) was only trend associated with the prognosis in OSCC. 

Interestingly, the amount of secreted CAIX protein correlated significantly with the level of secreted miRNA-210 in the same blood. Both markers are regulated by HIF1 [30,72] and possibly released in similar amounts by the tumor cells. The CA9 mRNA and the miRNA-210 levels determined in the same tumor tissue were correlated. However, this miRNA-210 concentration did not correlate with the CAIX protein in the tumor tissue, suggesting a posttranscriptional regulation of CAIX protein content (Table 3).

This analysis showed the prognostic value of the CA9 mRNA or CAIX protein from the tumor tissue. However, the secreted CAIX protein levels measured from the blood serum of OSCC patients of the same cohort do not appear to be a significant prognostic marker.

## 4. Materials and Methods

### 4.1. Tissue Samples, Histomorphological Data and Study Approval

Seventy-two fresh frozen OSCC samples, as well as 24 fresh frozen tumor-adjacent tissue samples, had been used for real-time quantitative PCR. In a retrospective cohort of 158 OSCC samples, an immunohistochemical (IHC) staining for CAIX was performed and 89 serum blood samples were analyzed by ELISA. The analyzed mRNA samples isolated from fresh material from 37 patients are identical to the tumor tissue analyzed by the immunohistochemistry (IHC) of formalin-fixed tissue at the CAIX protein level. For 39 patients, the CA9 mRNA level in tumor tissue was analyzed, and for the identical patients, the CAIX protein level in the patient’s serum could also be examined (Table 3). From 15 patients, both the formalin-fixed tissue as well blood serum was analyzed.

All patients had been treated with surgery at the Department of Oral and Maxillofacial Plastic Surgery, Martin Luther University Halle Wittenberg, Germany. The tissue samples for mRNA/miRNA analysis were cut with a cryocut microtome and the first and the last histologic sections were stained with hematoxylin and eosin. Experienced pathologists (DB, UB) verified the sections. We defined samples as tumor tissue when >70% of the first and the last histologic sections were tumor tissue. The clinical and histomorphological parameters of OSCC patients are shown in Table 1 and Table 2. The study was carried out in compliance with the Helsinki Declaration, and it was approved by the Ethics Committee of the Medical Faculty of the University Halle (Ethic number: 210/19.08.09/10 and 2017-81 issued on 27 June 2017). All patients gave written informed consent (Department of Oral and Maxillofacial Plastic Surgery, University of Halle-Wittenberg, Germany).

### 4.2. RNA-Isolation

For this analysis, frozen tumor material was used from patients who underwent surgery from 2009–2012 in the Department of Oral and Maxillofacial Plastic Surgery of Martin Luther University, Halle, Wittenberg. From 24 patients, we analyzed fresh, frozen tumor-adjacent tissue. These patients also gave written informed consent because of the suspicion of a tumor in the oral cavity. In these 24 patient samples, however, the histological analysis of existing tissue biopsy by experienced pathologists (DB, UB) did not detect tumor tissue and they were classified as normal tissue.

Snap-frozen tumor samples were cut into 20 µm tissue sections and RNA and miRNA was isolated by Trizol reagent according to the manufacturer’s protocol (Invitrogen, Karlsruhe, Germany). DNA contaminations were removed by DNAse I digestion (Qiagen, Hilden, Germany). The RNA concentration was determined using a Nanodrop spectrophotometer (Thermo Scientific, Karlsruhe, Germany). RNA from 250 µL serum blood from patients was isolated by miRCURY™ RNA Isolation Kit–Biofluids from Exiqon (Vedbæk; Denmark) according to the manufacturer’s protocol.

### 4.3. Quantitative RT-PCR

Six µg of total RNA was used for cDNA synthesis (tissue samples) according to standard protocols (Fermentas, St. Leon-Rot, Germany) with random hexamer primers in a Thermo-Trioblock TB1 (Biometra, Göttingen, Germany) for one hour at 42 °C as previously described [53,64]. The cDNA for the miRNA analysis was synthesized with the miRCURY LNA™ Universal RT microRNA PCR–Universal cDNA synthesis kit II Exiqon (Vedbæk; Denmark) according to the manufacturer’s protocol. Briefly, 10 µL of a reaction mixture comprising of 50 ng RNA (isolated tumor tissue) or 5 ng RNA (isolated from serum blood) and a synthetic spike in RNA were incubated for one hour at 42 °C in primers in a Thermo-Trioblock TB1 (Biometra, Göttingen, Germany).

The cDNA was amplified by automated real-time quantitative TaqManTM assays for CA9, KRT13, GAPDH, CDH, ZEB2, survivin and RPII as a reference gene using kits from Thermo Fisher Scientific (Darmstadt, Germany). mRNAs gene transcript amounts were normalized to RPII transcript amounts using the ΔΔ*C*t method [73]. Cycling conditions consisted of a single activation step at 95 °C for 10 min followed by 40 cycles at 95 °C for 10 s and 60 °C for 60 s. The cDNA of miRNA was amplified by a Kit from Exiqon (miRCURY LNA™ Universal RT microRNA PCR – ExiLENT SYBR® Green master mix) using the specific Primer for miRNA-210 designed by Exiqon (Vedbæk; Denmark). Cycling conditions consisted of a single activation step at 95 °C for 10 min followed by 40 cycles at 95 °C for 10 s and 60 °C for 60 s. The reference miRNA-103a for normalization was determined as the best reference miRNA out of 25 measured miRNA by using the normfinder software (https://moma.dk/normfinder-software). Samples were run on a RotorGene real-time-PCR cycler (LTF Labortechnik, Wasserburg, Germany) using the Rotor-Gene 6000 series Software 1.7.87. 

### 4.4. Immunohistochemistry

In a retrospective cohort of 158 OSCC samples, immunohistochemical (IHC) staining for CAIX was performed and the data were correlated with clinical and pathological records of patients (clinical and pathological data Table 1). All patients had been treated with surgery at the Department of Oral and Maxillofacial Plastic Surgery, Martin Luther University, Halle, Wittenberg, Germany from 1997-2015.

A total of 158 OSCC patient samples were stained for CAIX using mAB M75 (BioScience, Slovak Republic). Short tissue samples was deparaffinized, than slides were steamed with a preheated T-EDTA buffer (ZUC029-500, 1:10 dissolved, Zytomed Systems) (Berlin, Germany) at pH 6.0 and 98 °C for 30 min in an oven (Braun, type 3216), then rinsed with aqua dest and blocked for 7-10 minutes with 3% H_2_O_2_. Following another rinsing step and application of washing buffer (ZUC202-2500, 1:20 solution), Zytochem Plus HRP Kit/Plus Polymer System, Zytomed the CAIX mAb at a dilution of 1:200 was added dropwise on the tissue area and incubated for 30 minutes at room temperature (RT). Then the slides were incubated with a biotinylated secondary antibody (Broad Spectrum, Zytochem Plus HRP Kit, Zytomed) for 15 min at room temperature, rinsed with washing buffer followed by 15 min of incubation with horseradish peroxidase (HRP; Zytochem Plus HRP, Zytomed). The epitopes were visualized with DAB (10 min of DAB Substrate Kit, Zytomed). After further rinsing steps (aqua dest.), the slides were counterstained with hemalaun (Dr. K. Hollborn & Sons, Germany) for 30 s, rinsed in water for 10 min, then transferred via alcohol into xylol and finally cover-slipped for bright field analysis. Staining results were semi-quantitatively evaluated utilizing the immunoreactive score (IRS) as described by Remmele and Stegner [74]. The staining intensity (0—negative; 1—low; 2—medium; 3—strong positive) and the percentage of stained cells (0-negative, 1- <10%, 2-10-50%, 3- 51-80%, 4-> 80%) were evaluated, the immunoreactive score (IRS score) was then calculated as a product of the two parameters and ranged from 0–12. This work was performed by a pathologist (DB) via light field microscopy. The scores were stated as 0–3 as negative or weak, IRS 4–6 as moderate and IRS 8–12 as strong staining.

### 4.5. ELISA

Eighty-nine serum blood samples were collected during surgery (biopsy of fresh frozen material). Simultaneously, the tumor samples frozen for mRNA isolation were harvested, as well as the blood was centrifuged as serum (serum Z/9 mL Sarstedt monovette) and EDTA plasma (KE/9 mL Sarstedt monovette) (Sarstedt AG &Co AG, Nümbrecht, Germany) and stored at −80 °C. 

In addition, blood serum was isolated from patients from the years 2010–2013, whereby these patients were operated in the Department of Oral and Maxillofacial Plastic Surgery of Martin Luther University Halle Wittenberg.

Moreover, the blood of 43 healthy blood donors was analysed. The ELISA (Human Carbonic Anhydrase IX Assay, R&D systems, Minneapolis, Minnesota, USA) was performed according to the manufacturer’s instructions. CAIX serum concentration was determined using the standard curve (clinical and pathological data Table 3). Briefly, 100 µL samples were incubated with 50 µL Assay buffer for 2 h at room temperature (RT) in a 96-well plate, and then the well was washed four times with a 400 µL wash buffer, incubated with a 200 µL conjugate buffer for 2 h at RT, washed with four times wash buffer, followed by an incubation with a 200 µL substrate buffer for 30 min at RT (in the dark). After washing, a 50 µL stop solution was added and the plate was measured by a plate reader (GENios TECAN, Männedorf, Switzerland) at 450 nm. The concentration of CAIX protein in each sample was calculated using a standard curve.

### 4.6. Statistical Analysis

The statistical analysis was univariately performed with Kaplan–Meier analysis to calculate overall and recurrence-free survival. Cox’s regression hazard model for analysis of overall survival and locoregional control was adjusted for the prognostic effect of covariates (T-stage and N-stage and tumor grade), and the relative risk (RR) was calculated. Survival times were calculated from the day of tumor diagnosis. The endpoint for the overall survival analysis was the time of death of the patient. The endpoint for the locoregional control analysis was the first recurrence. The interrelationships between the different mRNA levels were tested with the Spearman’s rank correlation (rs, correlation coefficient). The correlation of the CAIX/CA9 mRNA level, T-stage, N-stage, grading and gender of the patients were tested with the Kruskal Wallis test [64]. Receiver operating characteristic (ROC) curves were used to test the specificity of the CA9 mRNA level as a marker for the differentiator of tumor or normal tissue. A probability (P) of <0.05 was defined as significant. Statistical analyses were carried out using SPSS software version 25.0 (IBM, Armonk, New York, USA).

## 5. Conclusions

To the best of our knowledge, this is the first paper demonstrating a significant association between the prognosis of OSCC patients and their levels of CA9 mRNA and CAIX protein in tumor tissue. Our results confirm that immunohistochemical (IHC) staining correlates with mRNA levels of CAIX. Moreover, the HIF-1 α pathway as a hallmark of OSCC and its target gene/protein CAIX could support future therapeutic strategies.

## Figures and Tables

**Figure 1 ijms-20-00375-f001:**
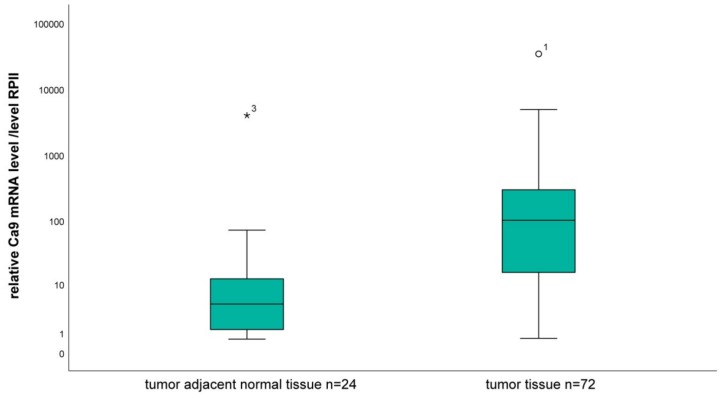
The relative CA9 mRNA level in tumor tissue samples from 72 oral squamous cell carcinoma patients compared to samples from 24 tumor-adjacent normal tissue. (* and ° extreme data points).

**Figure 2 ijms-20-00375-f002:**
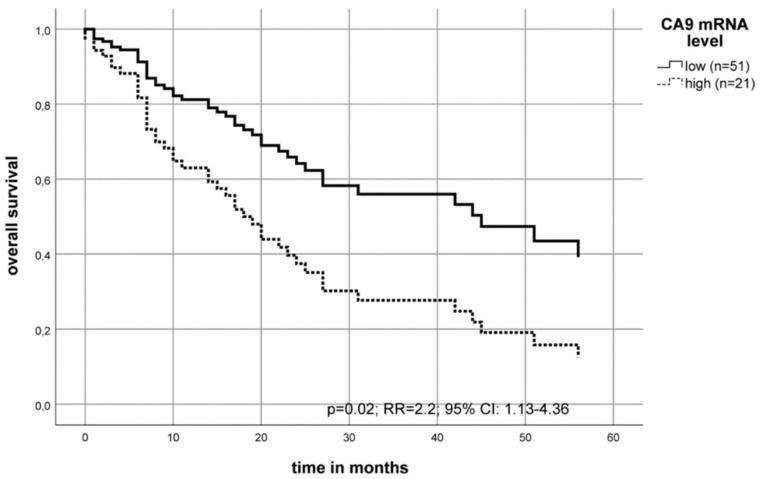
The multivariate Cox’s regression hazard model: Association of CA9 mRNA level and overall survival of OSCC patients. The model was adjusted to the patient’s tumor stage, lymph node status (N-stage) and tumor grade. The OSCC cohort was separated into two groups. The patient’s risk of death was calculated as RR = 2.2 (*p* = 0.02) for a high CA9 mRNA level compared to the group with a low level of CA9 mRNA.

**Figure 3 ijms-20-00375-f003:**
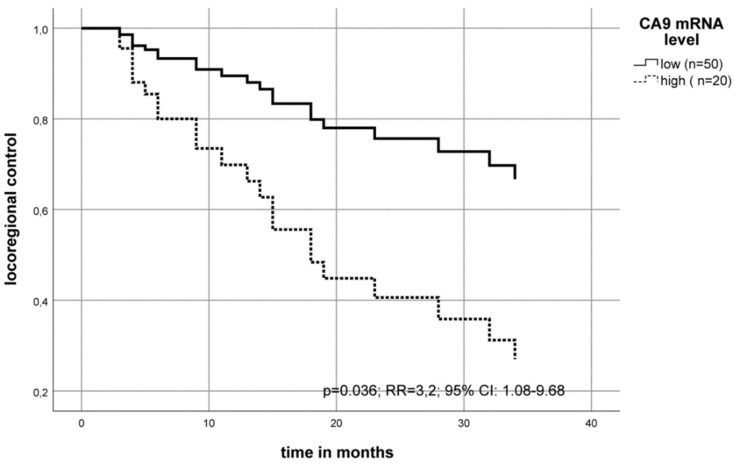
The multivariate Cox’s regression hazard model: Association of CA9 mRNA expression levels and locoregional recurrence of OSCC patients. The model was adjusted to the patient’s tumor stage, lymph node status (N-stage) and tumor grade. The OSCC cohort was separated into two groups according to the CA9 mRNA levels. The risk of recurrence was calculated as RR = 3.2 (*p* = 0.036) for a high CA9 mRNA expression compared to the group with a low level of CA9 mRNA.

**Figure 4 ijms-20-00375-f004:**
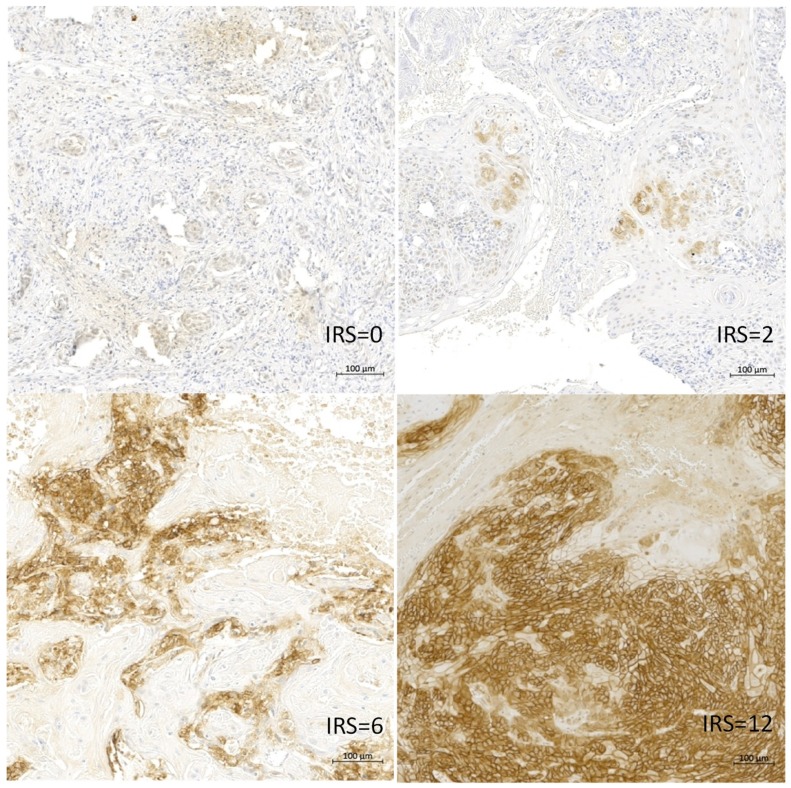
The examples of an Immunohistochemistry (IHC) staining for the CAIX protein in the tumor tissue of OSCC for samples with an immunoreactive score (IRS) of 0, 2, 6 and 12.

**Figure 5 ijms-20-00375-f005:**
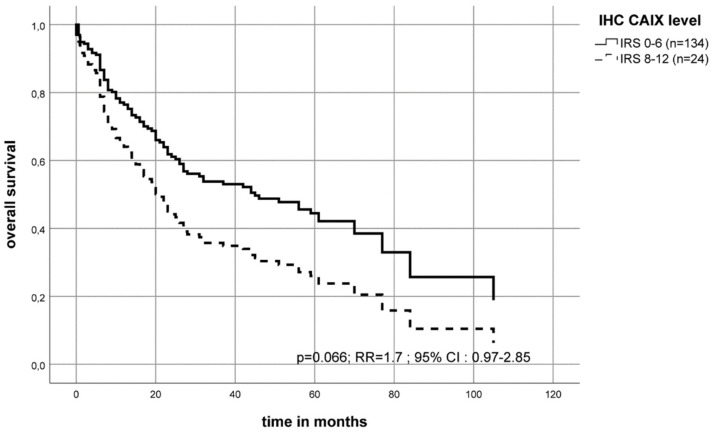
The multivariate Cox’s regression hazard model: Immunohistochemistry (IHC) staining of CAIX protein in tumor tissue. Association of CAIX protein level and survival of OSCC patients. The model was adjusted to the patient’s tumor stage, lymph node status (N-stage) and tumor grade. The OSCC cohort was separated into two groups. The patient`s risk of death was calculated as RR = 1.7 (*p* = 0.066) for a high CAIX protein level compared to the group with a low/moderate level of CAIX protein.

**Figure 6 ijms-20-00375-f006:**
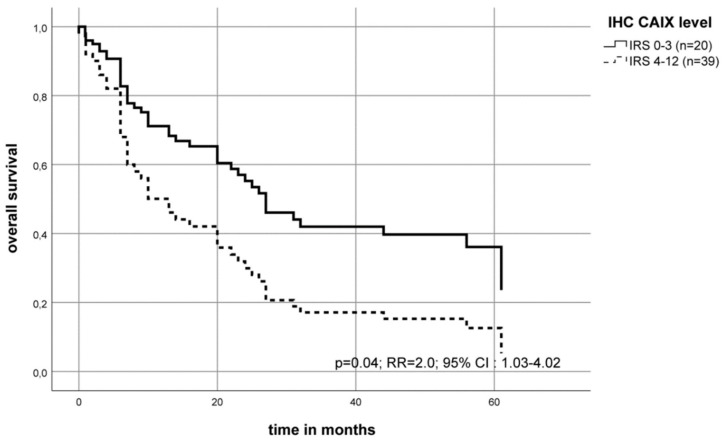
The multivariate Cox’s regression hazard model: Immunohistochemistry (IHC) staining for CAIX protein of the tumor tissue of T4 tumors. Association of the CAIX protein level and survival of OSCC patients. The model was adjusted to the patient`s lymph node status (N-stage) and tumorgrade. The OSCC cohort was separated into two groups. The patient`s risk of death was calculated as RR = 2.0 (*p* = 0.04) for a higher CAIX protein level (IRS 4–12) compared to the group with a negative/weak level (IRS 0–3).

**Figure 7 ijms-20-00375-f007:**
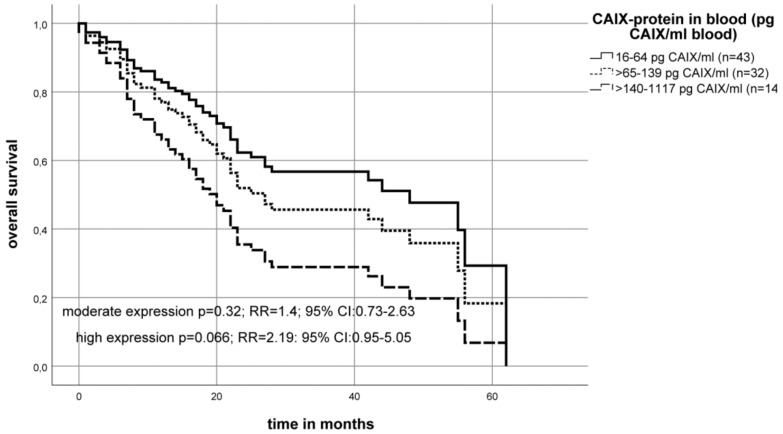
The multivariate Cox’s regression hazard model: Association of soluble CAIX protein level analyzed from patient serum and survival of OSCC patients. The model was adjusted to the patient’s tumor stage, lymph node status (N-stage) and tumor grade. The OSCC cohort was separated into three groups according to the serum CAIX protein level. The non-significant patient’s risk of death was calculated as RR = 1.4 (*p* = 0.32) for an expression of ≥ 65–139 pg CAIX/mL serum (moderate expression) and RR = 2.2 (*p* = 0.066) for an expression of ≥ 140–1117 pg CAIX/mL serum (high expression) compared to the control group (16–64 pg CAIX/mL serum) (low expression).

**Table 1 ijms-20-00375-t001:** The clinicopathological data of oral squamous cell carcinoma patients (mRNA and protein (IHC) data). *P*-value calculated using the Kruskal Wallis test. (*-significant result).

Category	Number of Cases (CA9)	Low CA9 mRNA Level (0.63–246)	High CA9 mRNA Level (246.1–33779)	Number of Cases (CAIX)	Low CAIX Protein Level (IRS 0–6)	High CAIX Protein Level (IRS 8–12)
**Total**	72	51	21	158	134	24
**Gender**			*p* = 1.00			*p* = 0.29
men	58	41	17	123	102	21
women	14	10	4	35	32	3
**T-stage**			*p* = 0.22			***p* = 0.03 ***
I	12	10	2	37	35	2
II	23	18	5	43	37	6
III	7	3	4	19	16	3
IV	30	20	10	59	46	13
**N-stage**			*p* = 0.60			*p* = 0.16
N0	28	21	7	72	64	8
N1-3	44	30	14	86	70	16
**Grade**			*p* = 0.62			*p* = 0.54
1	9	8	1	16	15	1
2	52	35	17	93	78	15
3	10	7	3	49	41	8
*x*	1	1	0			

**Table 2 ijms-20-00375-t002:** The clinicopathological data of OSCC patients (blood CAIX (soluble CAIX) level) *p*-value calculated using the Kruskal Wallis test.

Category	Number of Cases	Low Soluble *CAIX* Protein Level	High Soluble CAIX Protein Level	Very High Soluble CAIX Protein Level
***Total***	89	43	32	14
***Gender***				*p* = 0.71
men	67	31	26	10
women	22	12	6	4
***T-stage***				*p* = 0.10
I	19	12	6	1
II	30	15	7	8
III	13	8	5	0
IV	27	8	14	5
***N-stage***				*p* = 0.70
*N0*	38	20	12	6
*N1-3*	51	23	20	8
***Grading***				*p* = 1.00
*1*	12	5	4	3
*2*	65	33	24	8
3	12	5	4	3

**Table 3 ijms-20-00375-t003:** The bivariate correlations between CA9 and CAIX (measured in the tumor and in the serum) of OSCC patients and different biomarkers (Spearman’s Rho test) (r_s_-correlation coefficient); TC—from the tumor cells.

	r_s_	*P*-value	n
**Ca9 mRNA in TC (n = 72) correlates with levels of**			
**hypoxic markers**			
*CAIX protein in TC*	0.611	**<0.001**	37
*miR-210 in TC*	0.331	**0.005**	70
*VEGFa mRNA in TC*	0.472	**<0.001**	72
*Glut1 mRNA in TC*	0.367	**0.002**	72
GAPDH mRNA *in TC*	−0.369	**0.001**	72
**structure proteins**			
*CDH mRNA in TC*	0.197	0.10	70
*KRT13 mRNA in TC*	0.230	**0.001**	69
**EMT and stem cell markers**			
*ZEB2-mRNA in TC*	−0.233	0.052	70
*Survivin-mRNA in TC*	0.371	**0.001**	72
**CAIX protein in TC (n = 158) correlates with levels of **			
**hypoxic markers**			
*CA9 mRNA in TC*	0.611	**<0.001**	37
*HIF1α protein in TC*	0.234	**0.015**	107
*miR-210 in TC*	0.054	0.75	36
**CAIX protein in blood serum (n = 89) correlates with levels of **			
**hypoxic markers**			
*CA9 mRNA in TC*	0.115	0.487	39
*CAIX protein in TC*	0.432	0.11	15
*miR-210 in TC*	−0.060	0.724	37
*miR-210 in serum*	0.457	**0.037**	21
**other markers**			
*Her2 mRNA in TC*	−0.339	**0.040**	37
*Her2 protein in TC*	0.263	0.204	25
Her2 protein in serum	0.300	0.064	39
*EGFR mRNA in TC*	−0.325	0.050	37

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
