# Peer review of "Investigation of the Prognostic Role of Carbonic Anhydrase 9 (CAIX) of the Cellular mRNA/Protein Level or Soluble CAIX Protein in Patients with Oral Squamous Cell Carcinoma"

_ijms, 2019, doi:10.3390/ijms20020375_

Round 1
Reviewer 1 Report
The authors present an analysis of the expression of carbonic anhydrase 9 in Oral Squamous Cell Carcinoma (OSCC) patients using three methods: Real-Time PCR for mRNA quantification in tumours, immunohistochemistry detection of the protein in deparaffinized tumour samples and ELISA quantification of secreted protein in the serum. A comparative analysis of these measurements was then carried out and their prognostic value was evaluated by examining survival and locoregional recurrence.
The main results show that: 1) mRNA expression levels are higher in tumour samples compared to adjacent normal tissue; 2) an increased risk of death and recurrence is associated with high mRNA levels and high protein levels in the tumour; 3) high protein levels in the serum are associated with a non-statistically significant tendency to an increased risk of death; 5) high mRNA and protein levels in the tumour did not correlate with secreted protein levels in the serum; 6) as expected, CA9 mRNA levels in the tumour were associated with those of mRNAs from other hypoxic markers.
These observations essentially confirm previous findings from the same group and from other groups suggesting that CA9 expression is a prognostic marker in OSCC, as recently reviewed by César Rivera et al (Oral Oncology 2017).
Although the amount of data presented is remarkable, I have several concerns that the authors should address.
First of all, the authors should clarify the novelty of their aims, in the Introduction, and the novelty of their findings, in the Discussion.
The Introduction should describe previous publications on CA9 expression in OSCC and mention similar data in other tumour types.
In line 48 the authors state that “the prognosis of OSCC has stagnated …therefore new prognostic molecular parameters are needed…”. Indeed, in 2018, a large body of publications on the subject is available and should be mentioned in the Introduction. Several new molecular markers have been proposed such as miRNAs in OSCC, telomeric repeat containing RNA in HNSCC, lncRNAs in HNSCC.
What is the origin of the samples tested with the different methods? It seems that the 89 serum samples were taken from some of the 158 patients analysed by IHC. Are the 72 samples analysed by RT-PCR related to the samples analysed by IHC and ELISA? This point should be clarified in the Results. In the Methods section the origin of all samples, including the adjacent normal tissues, should be clearly specified.
In the first paragraph of the Results, RT-PCR data should be presented in a graph showing all values of tumours and normal tissue samples, with median and mean values indicated in the graph. Mentioning median and mean values of the distributions in the text is not sufficient. In my opinion, such graph would be more informative than the ROC curve. The authors should clarify whether the 24 normal tissue samples were adjacent to 24 of the 72 tumours. If so, it would be interesting to show the mRNA levels in each tumour/adjacent tissue sample. Is the mRNA level in the tumour always higher than in its adjacent healthy tissue?
To evaluate protein expression by IHC arbitrary IRS values were used. A brief description of this quantification should be added to the Methods section. Were normal tissues analysed for comparison? What is their IRS value?
In the abstract the authors state that secreted protein levels were not significantly associated with prognosis while in the Results they state that multivariate Cox’s regression analysis revealed “a tendency to a significant increased risk of death”. This conclusion should be described more precisely.
All abbreviations should be given in full at first mention
Reviewer 2 Report
The manuscript by Eckert et al examined the prognostic value of carbonic anhydrase 9 (CAIX) at the protein and mRNA (CA9) expression level during oral squamous cell carcinoma (OSCC) tumor progression. The results identify the secreted form of CAIX protein levels were not significantly associated with tumor prognosis. However, show the CA9/CAIX protein levels correlated with each other and therefore may be a therapeutic target. However, the authors should address the following comments.
Title-should specify the findings ie “cellular CAIX or soluble or CA9 mRNA/CAIX cellular protein etc”
Introduction-written briefly. They should include other protein markers/and gene signatures known (ref#3, 18,19 etc) to have prognostic value in OSCC tumor growth/invasion to bone. It has been recently reported (ref#16) that CAIX over-expression regulates migration and progression of OSCC. Please clarify a rationale and new findings of the present study.
Table.3 (p.10) summarizes HIF1alpha protein expression, but not more stable form HIF1 which regulate CAIX gene expression. They should clarify the manuscript abstract, introduction and discussion sections for the rationale and results pertaining to correlation of HIF1 expression than HIF1 alpha.
Fig.4 legend should define the abbreviation IRS used in panels.
Materials and Methods- this section is noted after the Discussion. Please verify the journal policy or an article published.
Minor comments:
Fig.3 legend (p.5) “CA9 mRNA or Ca9 RNA “ used should be consistent throughout the text.
Correct the citation for Zhou ad co-workers (p.10, line 204) ref#24 or ref# 23.
Ref#16- Remove the extra words like “the journal of the International Society…Biology and Medicine”
Round 2
Reviewer 2 Report
None